# A Kernel Test for Three-Variable Interactions

**Dino Sejdinovic, Arthur Gretton**
Gatsby Unit, CSML, UCL, UK
{dino.sejdinovic, arthur.gretton}@gmail.com

**Wicher Bergsma**
Department of Statistics, LSE, UK
w.p.bergsma@lse.ac.uk

## Abstract

We introduce kernel nonparametric tests for Lancaster three-variable interaction and for total independence, using embeddings of signed measures into a reproducing kernel Hilbert space. The resulting test statistics are straightforward to compute, and are used in powerful interaction tests, which are consistent against all alternatives for a large family of reproducing kernels. We show the Lancaster test to be sensitive to cases where two independent causes individually have weak influence on a third dependent variable, but their combined effect has a strong influence. This makes the Lancaster test especially suited to finding structure in directed graphical models, where it outperforms competing nonparametric tests in detecting such V-structures.

## 1 Introduction

The problem of nonparametric testing of interaction between variables has been widely treated in the machine learning and statistics literature. Much of the work in this area focuses on measuring or testing pairwise interaction: for instance, the Hilbert-Schmidt Independence Criterion (HSIC) or Distance Covariance [1, 2, 3], kernel canonical correlation [4, 5, 6], and mutual information [7]. In cases where more than two variables interact, however, the questions we can ask about their interaction become significantly more involved. The simplest case we might consider is whether the variables are mutually independent, $P_X = \prod_{i=1}^{d} P_{X_i}$, as considered in $\mathbb{R}^d$ by [8]. This is already a more general question than pairwise independence, since pairwise independence does not imply total (mutual) independence, while the implication holds in the other direction. For example, if $X$ and $Y$ are i.i.d. uniform on $\{-1, 1\}$, then $(X, Y, XY)$ is a pairwise independent but mutually dependent triplet [9]. Tests of total and pairwise independence are insufficient, however, since they do not rule out all third order factorizations of the joint distribution.

An important class of high order interactions occurs when the simultaneous effect of two variables on a third may not be additive. In particular, it may be possible that $X \perp\!\!\!\perp Z$ and $Y \perp\!\!\!\perp Z$, whereas $\neg((X, Y) \perp\!\!\!\perp Z)$ (for example, neither adding sugar to coffee nor stirring the coffee individually have an effect on its sweetness but the joint presence of the two does). In addition, study of three-variable interactions can elucidate certain switching mechanisms between positive and negative correlation of two genes expressions, as controlled by a third gene [10]. The presence of such interactions is typically tested using some form of analysis of variance (ANOVA) model which includes additional interaction terms, such as products of individual variables. Since each such additional term requires a new hypothesis test, this increases the risk that some hypothesis test will produce a false positive by chance. Therefore, a test that is able to directly detect the presence of *any kind* of higher-order interaction would be of a broad interest in statistical modeling. In the present work, we provide to our knowledge the first nonparametric test for three-variable interaction. This work generalizes the HSIC test of pairwise independence, and has as its test statistic the norm

of an embedding of an appropriate signed measure to a reproducing kernel Hilbert space (RKHS). When the statistic is non-zero, all third order factorizations can be ruled out. Moreover, this test is applicable to the cases where $X$, $Y$ and $Z$ are themselves multivariate objects, and may take values in non-Euclidean or structured domains.[1]

One important application of interaction measures is in learning structure for graphical models. If the graphical model is assumed to be Gaussian, then second order interaction statistics may be used to construct an undirected graph [11, 12]. When the interactions are non-Gaussian, however, other approaches are brought to bear. An alternative approach to structure learning is to employ conditional independence tests. In the PC algorithm [13, 14, 15], a V-structure (a directed graphical model with two independent parents pointing to a single child) is detected when an independence test between the parent variables accepts the null hypothesis, while a test of dependence of the parents conditioned on the child rejects the null hypothesis. The PC algorithm gives a correct equivalence class of structures subject to the causal Markov and faithfulness assumptions, in the absence of hidden common causes. The original implementations of the PC algorithm rely on partial correlations for testing, and assume Gaussianity. A number of algorithms have since extended the basic PC algorithm to arbitrary probability distributions over multivariate random variables [16, 17, 18], by using nonparametric kernel independence tests [19] and conditional dependence tests [20, 18]. We observe that our Lancaster interaction based test provides a strong alternative to the conditional dependence testing approach, and is seen to outperform earlier approaches in detecting cases where independent parent variables weakly influence the child variable when considered individually, but have a strong combined influence.

We begin our presentation in Section 2 with a definition of interaction measures, these being the signed measures we will embed in an RKHS. We cover this embedding procedure in Section 3. We then proceed in Section 4 to define pairwise and three way interactions. We describe a statistic to test mutual independence for more than three variables, and provide a brief overview of the more complex high-order interactions that may be observed when four or more variables are considered. Finally, we provide experimental benchmarks in Section 5.

## 2  Interaction Measure

An interaction measure [21, 22] associated to a multidimensional probability distribution $P$ of a random vector $(X_1, \ldots, X_D)$ taking values in the product space $\mathcal{X}_1 \times \cdots \times \mathcal{X}_D$ is a signed measure $\Delta P$ that vanishes whenever $P$ can be factorised in a non-trivial way as a product of its (possibly multivariate) marginal distributions. For the cases $D = 2, 3$ the correct interaction measure coincides with the the notion introduced by Lancaster [21] as a formal product

$$\Delta_L P \;\; = \;\; \prod_{i=1}^{D} \left( P_{X_i}^* - P_{X_i} \right), \tag{1}$$

where each product $\prod_{j=1}^{D'} P_{X_{i_j}}^*$ signifies the joint probability distribution $P_{X_{i_1} \cdots X_{i_{D'}}}$ of a subvector $\left( X_{i_1}, \ldots, X_{i_{D'}} \right)$. We will term the signed measure in (1) the *Lancaster interaction measure*. In the case of a bivariate distribution, the Lancaster interaction measure is simply the difference between the joint probability distribution and the product of the marginal distributions (the only possible non-trivial factorization for $D = 2$), $\Delta_L P = P_{XY} - P_X P_Y$, while in the case $D = 3$, we obtain

$$\Delta_L P \;\; = \;\; P_{XYZ} - P_{XY} P_Z - P_{YZ} P_X - P_{XZ} P_Y + 2 P_X P_Y P_Z. \tag{2}$$

It is readily checked that

$$(X, Y) \perp\!\!\!\perp Z \;\vee\; (X, Z) \perp\!\!\!\perp Y \;\vee\; (Y, Z) \perp\!\!\!\perp X \;\;\; \Rightarrow \;\;\; \Delta_L P = 0. \tag{3}$$

For $D > 3$, however, (1) does not capture all possible factorizations of the joint distribution, e.g., for $D = 4$, it need not vanish if $(X_1, X_2) \perp\!\!\!\perp (X_3, X_4)$, but $X_1$ and $X_2$ are dependent and $X_3$ and $X_4$ are dependent. Streitberg [22] corrected this definition using a more complicated construction with the Möbius function on the lattice of partitions, which we describe in Section 4.3. In this

work, however, we will focus on the case of three variables and formulate interaction tests based on embedding of (2) into an RKHS.

The implication (3) states that the presence of Lancaster interaction rules out the possibility of any factorization of the joint distribution, but the converse is not generally true; see Appendix C for details. In addition, it is important to note the distinction between the absence of Lancaster interaction and the total (mutual) independence of $(X, Y, Z)$, i.e., $P_{XYZ} = P_X P_Y P_Z$. While total independence implies the absence of Lancaster interaction, the signed measure $\Delta_{tot} P = P_{XYZ} - P_X P_Y P_Z$ associated to the total (mutual) independence of $(X, Y, Z)$ does not vanish if, e.g., $(X, Y) \perp\!\!\!\perp Z$, but $X$ and $Y$ are dependent.

In this contribution, we construct the non-parametric test for the hypothesis $\Delta_L P = 0$ (no Lancaster interaction), as well as the non-parametric test for the hypothesis $\Delta_{tot} P = 0$ (total independence), based on the embeddings of the corresponding signed measures $\Delta_L P$ and $\Delta_{tot} P$ into an RKHS. Both tests are particularly suited to the cases where $X$, $Y$ and $Z$ take values in a high-dimensional space, and, moreover, they remain valid for a variety of non-Euclidean and structured domains, i.e., for all topological spaces where it is possible to construct a valid positive definite function; see [23] for details. In the case of total independence testing, our approach can be viewed as a generalization of the tests proposed in [24] based on the empirical characteristic functions.

## 3    Kernel Embeddings

We review the embedding of signed measures to a reproducing kernel Hilbert space. The RKHS norms of such embeddings will then serve as our test statistics. Let $\mathcal{Z}$ be a topological space. According to the Moore-Aronszajn theorem [25, p. 19], for every symmetric, positive definite function (henceforth *kernel*) $k : \mathcal{Z} \times \mathcal{Z} \to \mathbb{R}$, there is an associated reproducing kernel Hilbert space (RKHS) $\mathcal{H}_k$ of real-valued functions on $\mathcal{Z}$ with reproducing kernel $k$. The map $\varphi : \mathcal{Z} \to \mathcal{H}_k$, $\varphi : z \mapsto k(\cdot, z)$ is called the canonical feature map or the Aronszajn map of $k$. Denote by $\mathcal{M}(\mathcal{Z})$ the Banach space of all finite signed Borel measures on $\mathcal{Z}$. The notion of a feature map can then be extended to kernel embeddings of elements of $\mathcal{M}(\mathcal{Z})$ [25, Chapter 4].

**Definition 1.** (**Kernel embedding**) Let $k$ be a kernel on $\mathcal{Z}$, and $\nu \in \mathcal{M}(\mathcal{Z})$. The *kernel embedding* of $\nu$ into the RKHS $\mathcal{H}_k$ is $\mu_k(\nu) \in \mathcal{H}_k$ such that $\int f(z) d\nu(z) = \langle f, \mu_k(\nu) \rangle_{\mathcal{H}_k}$ for all $f \in \mathcal{H}_k$.

Alternatively, the kernel embedding can be defined by the Bochner integral $\mu_k(\nu) = \int k(\cdot, z) \, d\nu(z)$. If a measurable kernel $k$ is a bounded function, it is straightforward to show using the Riesz representation theorem that $\mu_k(\nu)$ exists for all $\nu \in \mathcal{M}(\mathcal{Z})$.[2] For many interesting bounded kernels $k$, including the Gaussian, Laplacian and inverse multiquadratics, the embedding $\mu_k : \mathcal{M}(\mathcal{Z}) \to \mathcal{H}_k$ is injective. Such kernels are said to be *integrally strictly positive definite* (ISPD) [26, p. 4]. A related but weaker notion is that of a *characteristic* kernel [20, 27], which requires the kernel embedding to be injective only on the set $\mathcal{M}_+^1(\mathcal{Z})$ of probability measures. In the case that $k$ is ISPD, since $\mathcal{H}_k$ is a Hilbert space, we can introduce a notion of an inner product between two signed measures $\nu, \nu' \in \mathcal{M}(\mathcal{Z})$,

$$\langle\langle \nu, \nu' \rangle\rangle_k := \langle \mu_k(\nu), \mu_k(\nu') \rangle_{\mathcal{H}_k} = \int k(z, z') d\nu(z) d\nu'(z').$$

Since $\mu_k$ is injective, this is a valid inner product and induces a norm on $\mathcal{M}(\mathcal{Z})$, for which $\|\nu\|_k = \langle\langle \nu, \nu \rangle\rangle_k^{1/2} = 0$ if and only if $\nu = 0$. This fact has been used extensively in the literature to formulate: (a) a nonparametric two-sample test based on estimation of *maximum mean discrepancy* $\|P - Q\|_k$, for samples $\{X_i\}_{i=1}^n \overset{i.i.d.}{\sim} P$, $\{Y_i\}_{i=1}^m \overset{i.i.d.}{\sim} Q$ [28] and (b) a nonparametric independence test based on estimation of $\|P_{XY} - P_X P_Y\|_{k \otimes l}$, for a joint sample $\{(X_i, Y_i)\}_{i=1}^n \overset{i.i.d.}{\sim} P_{XY}$ [19] (the latter is also called a Hilbert-Schmidt independence criterion), with kernel $k \otimes l$ on the product space defined as $k(x, x')l(y, y')$. When a bounded characteristic kernel is used, the above tests are *consistent against all alternatives*, and their alternative interpretation is as a generalization [29, 3] of energy distance [30, 31] and distance covariance [2, 32].

Table 1: $V$-statistic estimates of $\langle\langle \nu, \nu'\rangle\rangle_{k\otimes l}$ in the two-variable case

| $\nu \backslash \nu'$ | $P_{XY}$ | $P_X P_Y$ |
|---|---|---|
| $P_{XY}$ | $\frac{1}{n^2}(K \circ L)_{++}$ | $\frac{1}{n^3}(KL)_{++}$ |
| $P_X P_Y$ | | $\frac{1}{n^4}K_{++}L_{++}$ |

In this article, we extend this approach to the three-variable case, and formulate tests for both the Lancaster interaction and for the total independence, using simple consistent estimators of $\|\Delta_L P\|_{k\otimes l\otimes m}$ and $\|\Delta_{tot} P\|_{k\otimes l\otimes m}$ respectively, which we describe in the next Section. Using the same arguments as in the tests of [28, 19], these tests are also consistent against all alternatives as long as ISPD kernels are used.

## 4  Interaction Tests

**Notational remarks:** Throughout the paper, $\circ$ denotes an Hadamard (entrywise) product. Let $A$ be an $n \times n$ matrix, and $K$ a symmetric $n \times n$ matrix. We will fix the following notational conventions: $\mathbf{1}$ denotes an $n \times 1$ column of ones; $A_{+j} = \sum_{i=1}^{n} A_{ij}$ denotes the sum of all elements of the $j$-th column of $A$; $A_{i+} = \sum_{j=1}^{n} A_{ij}$ denotes the sum of all elements of the $i$-th row of $A$; $A_{++} = \sum_{i=1}^{n}\sum_{j=1}^{n} A_{ij}$ denotes the sum of all elements of $A$; $K_+ = \mathbf{1}\mathbf{1}^\top K$, i.e., $[K_+]_{ij} = K_{+j} = K_{j+}$, and $\left[K_+^\top\right]_{ij} = K_{i+} = K_{+i}$.

### 4.1  Two-Variable (Independence) Test

We provide a short overview of the kernel independence test of [19], which we write as the RKHS norm of the embedding of a signed measure. While this material is not new (it appears in [28, Section 7.4]), it will help define how to proceed when a third variable is introduced, and the signed measures become more involved. We begin by expanding the squared RKHS norm $\|P_{XY} - P_X P_Y\|_{k\otimes l}^2$ as inner products, and applying the reproducing property,

$$\begin{aligned}
\|P_{XY} - P_X P_Y\|_{k\otimes l}^2 &= \mathbb{E}_{XY}\mathbb{E}_{X'Y'}k(X,X')l(Y,Y') + \mathbb{E}_X\mathbb{E}_{X'}k(X,X')\mathbb{E}_Y\mathbb{E}_{Y'}l(Y,Y') \\
&\quad - 2\mathbb{E}_{X'Y'}\left[\mathbb{E}_X k(X,X')\mathbb{E}_Y l(Y,Y')\right],
\end{aligned} \tag{4}$$

where $(X,Y)$ and $(X',Y')$ are independent copies of random variables on $\mathcal{X} \times \mathcal{Y}$ with distribution $P_{XY}$.

Given a joint sample $\{(X_i, Y_i)\}_{i=1}^{n} \overset{i.i.d.}{\sim} P_{XY}$, an empirical estimator of $\|P_{XY} - P_X P_Y\|_{k\otimes l}^2$ is obtained by substituting corresponding empirical means into (4), which can be represented using Gram matrices $K$ and $L$ ($K_{ij} = k(X_i, X_j)$, $L_{ij} = l(Y_i, Y_j)$),

$$\hat{\mathbb{E}}_{XY}\hat{\mathbb{E}}_{X'Y'}k(X,X')l(Y,Y') = \frac{1}{n^2}\sum_{a=1}^{n}\sum_{b=1}^{n} K_{ab}L_{ab} = \frac{1}{n^2}(K \circ L)_{++},$$

$$\hat{\mathbb{E}}_X\hat{\mathbb{E}}_{X'}k(X,X')\hat{\mathbb{E}}_Y\hat{\mathbb{E}}_{Y'}l(Y,Y') = \frac{1}{n^4}\sum_{a=1}^{n}\sum_{b=1}^{n}\sum_{c=1}^{n}\sum_{d=1}^{n} K_{ab}L_{cd} = \frac{1}{n^4}K_{++}L_{++},$$

$$\hat{\mathbb{E}}_{X'Y'}\left[\hat{\mathbb{E}}_X k(X,X')\hat{\mathbb{E}}_Y l(Y,Y')\right] = \frac{1}{n^3}\sum_{a=1}^{n}\sum_{b=1}^{n}\sum_{c=1}^{n} K_{ac}L_{bc} = \frac{1}{n^3}(KL)_{++}.$$

Since these are V-statistics [33, Ch. 5], there is a bias of $O_P(n^{-1})$; U-statistics may be used if an unbiased estimate is needed. Each of the terms above corresponds to an estimate of an inner product $\langle\langle \nu, \nu'\rangle\rangle_{k\otimes l}$ for probability measures $\nu$ and $\nu'$ taking values in $\{P_{XY}, P_X P_Y\}$, as summarized in Table 1. Even though the second and third terms involve triple and quadruple sums, each of the empirical means can be computed using sums of all terms of certain matrices, where the dominant computational cost is in computing the matrix product $KL$. In fact, the overall estimator can be

Table 2: $V$-statistic estimates of $\langle\langle \nu, \nu' \rangle\rangle_{k\otimes l\otimes m}$ in the three-variable case

| $\nu\backslash\nu'$ | $nP_{XYZ}$ | $n^2 P_{XY}P_Z$ | $n^2 P_{XZ}P_Y$ | $n^2 P_{YZ}P_X$ | $n^3 P_X P_Y P_Z$ |
|---|---|---|---|---|---|
| $nP_{XYZ}$ | $(K\circ L\circ M)_{++}$ | $((K\circ L)M)_{++}$ | $((K\circ M)L)_{++}$ | $((M\circ L)K)_{++}$ | $tr(K_+\circ L_+\circ M_+)$ |
| $n^2 P_{XY}P_Z$ | | $(K\circ L)_{++}M_{++}$ | $(MKL)_{++}$ | $(KLM)_{++}$ | $(KL)_{++}M_{++}$ |
| $n^2 P_{XZ}P_Y$ | | | $(K\circ M)_{++}L_{++}$ | $(KML)_{++}$ | $(KM)_{++}L_{++}$ |
| $n^2 P_{YZ}P_X$ | | | | $(L\circ M)_{++}K_{++}$ | $(LM)_{++}K_{++}$ |
| $n^3 P_X P_Y P_Z$ | | | | | $K_{++}L_{++}M_{++}$ |

computed in an even simpler form (see Proposition 9 in Appendix F), as $\left\|\hat{P}_{XY} - \hat{P}_X\hat{P}_Y\right\|^2_{k\otimes l} = \frac{1}{n^2}\left(K\circ HLH\right)_{++}$, where $H = I - \frac{1}{n}\mathbf{1}\mathbf{1}^\top$ is the centering matrix. Note that by the idempotence of $H$, we also have that $\left(K\circ HLH\right)_{++} = \left(HKH\circ HLH\right)_{++}$. In the rest of the paper, for any Gram matrix $K$, we will denote its corresponding centered matrix $HKH$ by $\tilde{K}$. When three variables are present, a two-variable test already allows us to determine whether for instance $(X,Y) \perp\!\!\!\perp Z$, i.e., whether $P_{XYZ} = P_{XY}P_Z$. It is sufficient to treat $(X,Y)$ as a single variable on the product space $\mathcal{X}\times\mathcal{Y}$, with the product kernel $k\otimes l$. Then, the Gram matrix associated to $(X,Y)$ is simply $K\circ L$, and the corresponding $V$-statistic is $\frac{1}{n^2}\left(K\circ L\circ \tilde{M}\right)_{++}$.[3] What is not obvious, however, is if a $V$-statistic for the Lancaster interaction (which can be thought of as a surrogate for the composite hypothesis of various factorizations) can be obtained in a similar form. We will address this question in the next section.

### 4.2 Three-Variable Tests

As in the two-variable case, it suffices to derive V-statistics for inner products $\langle\langle \nu, \nu' \rangle\rangle_{k\otimes l\otimes m}$, where $\nu$ and $\nu'$ take values in all possible combinations of the joint and the products of the marginals, i.e., $P_{XYZ}$, $P_{XY}P_Z$, etc. Again, it is easy to see that these can be expressed as certain expectations of kernel functions, and thereby can be calculated by an appropriate manipulation of the three Gram matrices. We summarize the resulting expressions in Table 2 - their derivation is a tedious but straightforward linear algebra exercise. For compactness, the appropriate normalizing terms are moved inside the measures considered.

Based on the individual RKHS inner product estimators, we can now easily derive estimators for various signed measures arising as linear combinations of $P_{XYZ}, P_{XY}P_Z$, and so on. The first such measure is an "incomplete" Lancaster interaction measure $\Delta_{(Z)}P = P_{XYZ}+P_X P_Y P_Z-P_{YZ}P_X-P_{XZ}P_Y$, which vanishes if $(Y,Z) \perp\!\!\!\perp X$ or $(X,Z) \perp\!\!\!\perp Y$, but not necessarily if $(X,Y) \perp\!\!\!\perp Z$. We obtain the following result for the empirical measure $\hat{P}$.

**Proposition 2** (Incomplete Lancaster interaction). $\left\|\Delta_{(Z)}\hat{P}\right\|^2_{k\otimes l\otimes m} = \frac{1}{n^2}\left(\tilde{K}\circ \tilde{L}\circ M\right)_{++}$.

Analogous expressions hold for $\Delta_{(X)}\hat{P}$ and $\Delta_{(Y)}\hat{P}$. Unlike in the two-variable case where either matrix or both can be centered, centering of each matrix in the three-variable case has a different meaning. In particular, one requires centering of all three kernel matrices to perform a "complete" Lancaster interaction test, as given by the following Proposition.

**Proposition 3** (Lancaster interaction). $\left\|\Delta_L\hat{P}\right\|^2_{k\otimes l\otimes m} = \frac{1}{n^2}\left(\tilde{K}\circ \tilde{L}\circ \tilde{M}\right)_{++}$.

The proofs of these Propositions are given in Appendix A. We summarize various hypotheses and the associated V-statistics in the Appendix B. As we will demonstrate in the experiments in Section 5, while particularly useful for testing the factorization hypothesis, i.e., for $(X,Y) \perp\!\!\!\perp Z \vee (X,Z) \perp\!\!\!\perp Y \vee (Y,Z) \perp\!\!\!\perp X$, the statistic $\left\|\Delta_L\hat{P}\right\|^2_{k\otimes l\otimes m}$ can also be used for powerful tests of either the individual hypotheses $(Y,Z) \perp\!\!\!\perp X$, $(X,Z) \perp\!\!\!\perp Y$, or $(X,Y) \perp\!\!\!\perp Z$, or for total independence testing,

i.e., $P_{XYZ} = P_X P_Y P_Z$, as it vanishes in all of these cases. The null distribution under each of these hypotheses can be estimated using a standard permutation-based approach described in Appendix D.

Another way to obtain the Lancaster interaction statistic is as the RKHS norm of the joint "central moment" $\Sigma_{XYZ} = \mathbb{E}_{XYZ}[(k_X - \mu_X) \otimes (l_Y - \mu_Y) \otimes (m_Z - \mu_Z)]$ of RKHS-valued random variables $k_X$, $l_Y$ and $m_Z$ (understood as an element of the tensor RKHS $\mathcal{H}_k \otimes \mathcal{H}_l \otimes \mathcal{H}_m$). This is related to a classical characterization of the Lancaster interaction [21, Ch. XII]: there is no Lancaster interaction between $X$, $Y$ and $Z$ if and only if $\mathrm{cov}\,[f(X), g(Y), h(Z)] = 0$ for all $L_2$ functions $f$, $g$ and $h$. There is an analogous result in our case (proof is given in Appendix A), which states

**Proposition 4.** $\|\Delta_L P\|_{k \otimes l \otimes m} = 0$ if and only if $\mathrm{cov}\,[f(X), g(Y), h(Z)] = 0$ for all $f \in \mathcal{H}_k$, $g \in \mathcal{H}_l$, $h \in \mathcal{H}_m$.

And finally, we give an estimator of the RKHS norm of the total independence measure $\Delta_{tot} P$.

**Proposition 5** (Total independence). *Let* $\Delta_{tot}\hat{P} = \hat{P}_{XYZ} - \hat{P}_X \hat{P}_Y \hat{P}_Z$. *Then:*

$$\left\| \Delta_{tot}\hat{P} \right\|_{k \otimes l \otimes m}^2 = \frac{1}{n^2} (K \circ L \circ M)_{++} - \frac{2}{n^4} tr(K_+ \circ L_+ \circ M_+) + \frac{1}{n^6} K_{++} L_{++} M_{++}.$$

The proof follows simply from reading off the corresponding inner-product V-statistics from the Table 2. While the test statistic for total independence has a somewhat more complicated form than that of Lancaster interaction, it can also be computed in quadratic time.

## 4.3 Interaction for $D > 3$

Streitberg's correction of the interaction measure for $D > 3$ has the form

$$\Delta_S P = \sum_\pi (-1)^{|\pi|-1} (|\pi| - 1)! J_\pi P, \tag{5}$$

where the sum is taken over all partitions of the set $\{1, 2, \ldots, n\}$, $|\pi|$ denotes the size of the partition (number of blocks), and $J_\pi : P \mapsto P_\pi$ is the *partition operator* on probability measures, which for a fixed partition $\pi = \pi_1 | \pi_2 | \ldots | \pi_r$ maps the probability measure $P$ to the product measure $P_\pi = \prod_{j=1}^r P_{\pi_j}$, where $P_{\pi_j}$ is the marginal distribution of the subvector $(X_i : i \in \pi_j)$. The coefficients correspond to the Möbius inversion on the partition lattice [34]. While the Lancaster interaction has an interpretation in terms of joint central moments, Streitberg's correction corresponds to joint cumulants [22, Section 4]. Therefore, a central moment expression like $\mathbb{E}_{X_1 \ldots X_n}[\left( k_{X_1}^{(1)} - \mu_{X_1} \right) \otimes \cdots \otimes \left( k_{X_n}^{(n)} - \mu_{X_n} \right)]$ does not capture the correct notion of the interaction measure. Thus, while one can in principle construct RKHS embeddings of higher-order interaction measures, and compute RKHS norms using a calculus of $V$-statistics and Gram-matrices analogous to that of Table 2, it does not seem possible to avoid summing over all partitions when computing the corresponding statistics, yielding a computationally prohibitive approach in general. This can be viewed by analogy with the scalar case, where it is well known that the second and third cumulants coincide with the second and third central moments, whereas the higher order cumulants are neither moments nor central moments, but some other polynomials of the moments.

## 4.4 Total independence for $D > 3$

In general, the test statistic for total independence in the $D$-variable case is

$$\left\| \hat{P}_{X_{1:D}} - \prod_{i=1}^D \hat{P}_{X_i} \right\|_{\otimes_{i=1}^D k^{(i)}}^2 = \frac{1}{n^2} \sum_{a=1}^n \sum_{b=1}^n \prod_{i=1}^D K_{ab}^{(i)} - \frac{2}{n^{D+1}} \sum_{a=1}^n \prod_{i=1}^D \sum_{b=1}^n K_{ab}^{(i)}$$

$$+ \frac{1}{n^{2D}} \prod_{i=1}^D \sum_{a=1}^n \sum_{b=1}^n K_{ab}^{(i)}.$$

A similar statistic for total independence is discussed by [24] where testing of total independence based on empirical characteristic functions is considered. Our test has a direct interpretation in terms of characteristic functions as well, which is straightforward to see in the case of translation invariant kernels on Euclidean spaces, using their Bochner representation, similarly as in [27, Corollary 4].

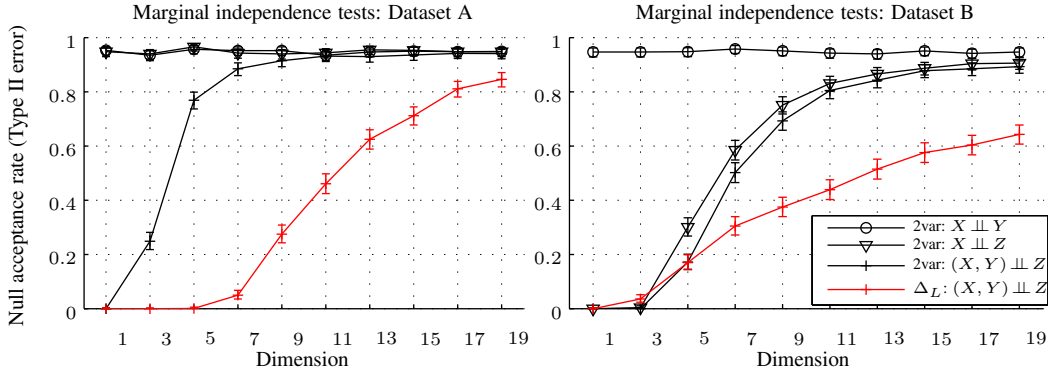

Figure 1: Two-variable kernel independence tests and the test for $(X, Y) \perp\!\!\!\perp Z$ using the Lancaster statistic

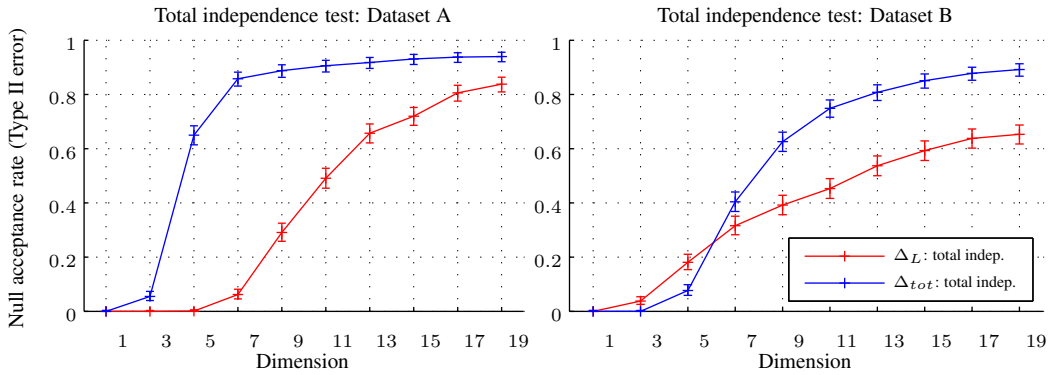

Figure 2: Total independence: $\Delta_{tot}\hat{P}$ vs. $\Delta_L\hat{P}$ .

## 5  Experiments

We investigate the performance of various permutation based tests that use the Lancaster statistic $\left\|\Delta_L \hat{P}\right\|^2_{k \otimes l \otimes m}$ and the total independence statistic $\left\|\Delta_{tot} \hat{P}\right\|^2_{k \otimes l \otimes m}$ on two synthetic datasets where $X, Y$ and $Z$ are random vectors of increasing dimensionality:

**Dataset A: Pairwise independent, mutually dependent data.** Our first dataset is a triplet of random vectors $(X, Y, Z)$ on $\mathbb{R}^p \times \mathbb{R}^p \times \mathbb{R}^p$, with $X, Y \overset{i.i.d.}{\sim} \mathcal{N}(0, I_p)$, $W \sim Exp(\frac{1}{\sqrt{2}})$, $Z_1 = sign(X_1 Y_1)W$, and $Z_{2:p} \sim \mathcal{N}(0, I_{p-1})$, i.e., the product of $X_1 Y_1$ determines the sign of $Z_1$, while the remaining $p - 1$ dimensions are independent (and serve as noise in this example).[4] In this case, $(X, Y, Z)$ is clearly a pairwise independent but mutually dependent triplet. The mutual dependence becomes increasingly difficult to detect as the dimensionality $p$ increases.

**Dataset B: Joint dependence can be easier to detect.** In this example, we consider a triplet of random vectors $(X, Y, Z)$ on $\mathbb{R}^p \times \mathbb{R}^p \times \mathbb{R}^p$, with $X, Y \overset{i.i.d.}{\sim} \mathcal{N}(0, I_p)$, $Z_{2:p} \sim \mathcal{N}(0, I_{p-1})$, and

$$Z_1 = \begin{cases} X_1^2 + \epsilon, & w.p. \ 1/3, \\ Y_1^2 + \epsilon, & w.p. \ 1/3, \\ X_1 Y_1 + \epsilon, & w.p. \ 1/3, \end{cases}$$

where $\epsilon \sim \mathcal{N}(0, 0.1^2)$. Thus, dependence of $Z$ on pair $(X, Y)$ is stronger than on $X$ and $Y$ individually.

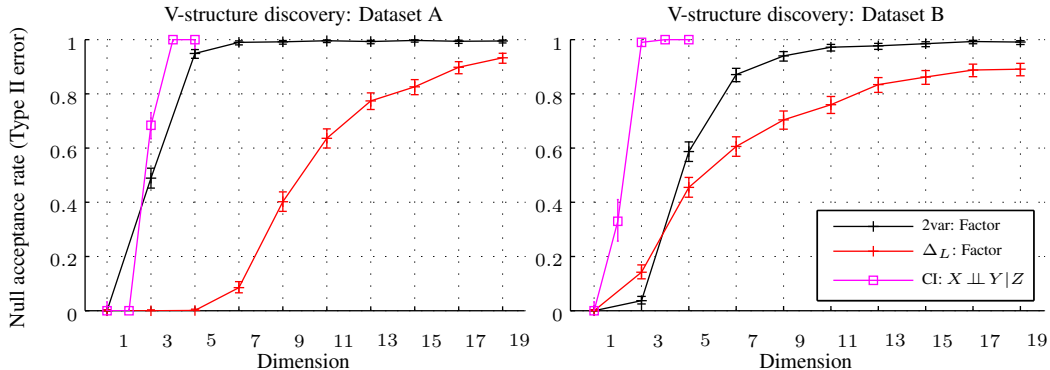

Figure 3: Factorization hypothesis: Lancaster statistic vs. a two-variable based test; Test for $X \perp\!\!\!\perp Y|Z$ from [18]

In all cases, we use permutation tests as described in Appendix D. The test level is set to $\alpha = 0.05$, sample size to $n = 500$, and we use gaussian kernels with bandwidth set to the interpoint median distance. In Figure 1, we plot the null hypothesis acceptance rates of the standard kernel two-variable tests for $X \perp\!\!\!\perp Y$ (which is true for both datasets A and B, and accepted at the correct rate across all dimensions) and for $X \perp\!\!\!\perp Z$ (which is true only for dataset A), as well as of the standard kernel two-variable test for $(X, Y) \perp\!\!\!\perp Z$, and the test for $(X, Y) \perp\!\!\!\perp Z$ using the Lancaster statistic. As expected, in dataset B, we see that dependence of $Z$ on pair $(X, Y)$ is somewhat easier to detect than on $X$ individually with two-variable tests. In both datasets, however, the Lancaster interaction appears significantly more sensitive in detecting this dependence as dimensionality $p$ increases. Figure 2 plots the Type II error of total independence tests with statistics $\left\| \Delta_L \hat{P} \right\|^2_{k \otimes l \otimes m}$ and $\left\| \Delta_{tot} \hat{P} \right\|^2_{k \otimes l \otimes m}$. The Lancaster statistic outperforms the total independence statistic everywhere apart from the Dataset B when the number of dimensions is small (between 1 and 5). Figure 3 plots the Type II error of the factorization test, i.e., test for $(X, Y) \perp\!\!\!\perp Z \vee (X, Z) \perp\!\!\!\perp Y \vee (Y, Z) \perp\!\!\!\perp X$ with Lancaster statistic with Holm-Bonferroni correction as described in Appendix D, as well as the two-variable based test (which performs three standard two-variable tests and applies the Holm-Bonferroni correction). We also plot the Type II error for the conditional independence test for $X \perp\!\!\!\perp Y|Z$ from [18]. Under assumption that $X \perp\!\!\!\perp Y$ (correct on both datasets), negation of each of these three hypotheses is equivalent to the presence of V-structure $X \rightarrow Z \leftarrow Y$, so the rejection of the null can be viewed as a V-structure detection procedure. As dimensionality increases, the Lancaster statistic appears significantly more sensitive to the interactions present than the competing approaches, which is particularly pronounced in Dataset A.

## 6    Conclusions

We have constructed permutation-based nonparametric tests for three-variable interactions, including the Lancaster interaction and total independence. The tests can be used in datasets where only higher-order interactions persist, i.e., variables are pairwise independent; as well as in cases where joint dependence may be easier to detect than pairwise dependence, for instance when the effect of two variables on a third is not additive. The flexibility of the framework of RKHS embeddings of signed measures allows us to consider variables that are themselves multidimensional. While the total independence case readily generalizes to more than three dimensions, the combinatorial nature of joint cumulants implies that detecting interactions of higher order requires significantly more costly computation.

## Footnotes

[1]As the reader might imagine, the situation becomes more complex again when four or more variables interact simultaneously; we provide a brief technical overview in Section 4.3.

[2]Unbounded kernels can also be considered, however [3]. In this case, one can still study embeddings of the signed measures $\mathcal{M}_k^{1/2}(\mathcal{Z}) \subset \mathcal{M}(\mathcal{Z})$, which satisfy a finite moment condition, i.e., $\mathcal{M}_k^{1/2}(\mathcal{Z}) = \left\{ \nu \in \mathcal{M}(\mathcal{Z}) : \int k^{1/2}(z, z) \, d|\nu|(z) < \infty \right\}$.

[3]In general, however, this approach would require some care since, e.g., $X$ and $Y$ could be measured on very different scales, and the choice of kernels $k$ and $l$ needs to take this into account.

[4]Note that there is no reason for $X, Y$ and $Z$ to have the same dimensionality $p$ - this is done for simplicity of exposition.

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
