[Supplementary Material]

# Appendix to "A Kernel Test for Three-Variable Interactions", *NIPS 2013*

**Dino Sejdinovic, Arthur Gretton, Wicher Bergsma**

## A   Proofs

### A.1   Proof of Proposition 2

Some basic matrix algebra used in this proof is reviewed in Appendix F. The proof of the following simple Lemma directly follows from the results therein.

**Lemma 6.** *The following equalities hold:*

1. $(K_+ \circ L_+ \circ M)_{++} = \left(K_+^\top \circ L_+^\top \circ M\right)_{++} = tr(K_+ \circ L_+ \circ M_+) = \sum_{a=1}^n K_{a+}L_{a+}M_{a+}$

2. $\left(K_+ \circ L \circ M_+^\top\right)_{++} = (KLM)_{++}$

Now, we will take a kernel matrix $M$ and consider its Hadamard product with $\tilde{K} \circ \tilde{L}$:

$$
\begin{aligned}
\tilde{K} \circ \tilde{L} \circ M \;=\;& K \circ L \circ M - \frac{1}{n}\Bigg[\underbrace{K \circ L_+ \circ M}_{A} + \underbrace{K \circ L_+^\top \circ M}_{A^\top} + \underbrace{K_+ \circ L \circ M}_{B} + \underbrace{K_+^\top \circ L \circ M}_{B^\top}\Bigg] \\
&+\; \frac{1}{n^2}\left(K_{++}L \circ M + L_{++}K \circ M\right) \\
&+\; \frac{1}{n^2}\Bigg[\underbrace{K_+ \circ L_+ \circ M}_{C} + \underbrace{K_+^\top \circ L_+^\top \circ M}_{C^\top} + \underbrace{K_+ \circ L_+^\top \circ M}_{D} + \underbrace{K_+^\top \circ L_+ \circ M}_{D^\top}\Bigg] \\
&-\; \frac{1}{n^3}K_{++}\left[L_+ \circ M + L_+^\top \circ M\right] - \frac{1}{n^3}L_{++}\left[K_+ \circ M + K_+^\top \circ M\right] \\
&+\; \frac{1}{n^4}K_{++}L_{++}M.
\end{aligned}
$$

and thus:

$$
\begin{aligned}
\left(\tilde{K} \circ \tilde{L} \circ M\right)_{++} \;=\;& (K \circ L \circ M)_{++} - \frac{2}{n}\left((K \circ M)L + (L \circ M)K\right)_{++} \\
&+\; \frac{1}{n^2}\left[K_{++}(L \circ M)_{++} + L_{++}(K \circ M)_{++}\right] \\
&+\; \frac{2}{n^2}\left[tr(K_+ \circ L_+ \circ M_+) + (LMK)_{++}\right] \\
&-\; \frac{2}{n^3}\left[K_{++}(LM)_{++} + L_{++}(KM)_{++}\right] \\
&+\; \frac{1}{n^4}K_{++}L_{++}M_{++}.
\end{aligned}
$$

where we used that $A_{++} = ((K \circ M) \circ L_+)_{++} = ((K \circ M)L)_{++}$, and similarly $B_{++} = ((L \circ M)K)_{++}$. Also, $C_{++} = tr(K_+ \circ L_+ \circ M_+)$ and $D_{++} = (LMK)_{++}$.

By comparing to the table of V-statistics, we obtain that:

$$
\frac{1}{n^2}\left(\tilde{K} \circ \tilde{L} \circ M\right)_{++} \;=\; \left\|\Delta_{(Z)}\hat{P}\right\|^2_{k \otimes l \otimes m}
$$

where $\Delta_{(Z)}\hat{P} = \hat{P}_{XYZ} + \hat{P}_X\hat{P}_Y\hat{P}_Z - \hat{P}_{YZ}\hat{P}_X - \hat{P}_{XZ}\hat{P}_Y$, which completes the proof of Proposition 2. Proposition 3 can be proved in an analogous way by including the additional terms corresponding to centering of $M$, i.e., $\left(\tilde{K} \circ \tilde{L} \circ M_+\right)_{++}$ and $\left(\tilde{K} \circ \tilde{L} \circ M_{++}\right)_{++}$. In the next Section, however, we give an alternative proof which gives more insight into the role that the centering of each Gram matrix plays.

## A.2 Proof of Proposition 3

It will be useful to introduce into notation the kernel centered at a probability measure $\nu$, given by:

$$\tilde{k}_\nu(z, z') := k(z, z') + \int\int k(w, w')d\nu(w)d\nu(w) - \int [k(z, w) + k(z', w)]\, d\nu(w), \qquad (6)$$

Note that $\int \tilde{k}_\nu(z, z')d\nu(z)d\nu(z') = 0$, i.e., $\mu_{\tilde{k}_\nu}(\nu) \equiv 0$.

By expanding the population expression of the kernel norm of the joint under the kernels centered at the marginals, we obtain:

$$\|P_{XYZ}\|^2_{\tilde{k}_{P_X} \otimes \tilde{l}_{P_Y} \otimes \tilde{m}_{P_Z}}$$

$$= \int\int \left[ \tilde{k}_{P_X}(x, x')\tilde{l}_{P_Y}(y, y')\tilde{m}_{P_Z}(z, z') \right]$$

$$dP_{XYZ}(x, y, z)dP_{XYZ}(x', y', z'),$$

Substituting the definition of the centered kernel in (6), it is readily obtained that

$$\|P_{XYZ}\|^2_{\tilde{k}_{P_X} \otimes \tilde{l}_{P_Y} \otimes \tilde{m}_{P_Z}} = \|\Delta_L P\|^2_{k\otimes l\otimes m}.$$

Now, $\|P_{XYZ}\|^2_{\tilde{k}_{P_X} \otimes \tilde{l}_{P_Y} \otimes \tilde{m}_{P_Z}}$ is the first term in the expansion of $\|\Delta_L P\|^2_{\tilde{k}_{P_X} \otimes \tilde{l}_{P_Y} \otimes \tilde{m}_{P_Z}}$. Let us show that all the other terms are equal to zero. Indeed, all the other terms are of the form

$$\langle\langle P_W Q, Q'\rangle\rangle_{\tilde{k}_{P_X} \otimes \tilde{l}_{P_Y} \otimes \tilde{m}_{P_Z}},$$

where $W = X, Y$, or $Z$ (individual variable). Without loss of generality, let $W = X$. Then,

$$\langle\langle P_X Q, Q'\rangle\rangle_{\tilde{k}_{P_X} \otimes \tilde{l}_{P_Y} \otimes \tilde{m}_{P_Z}}$$

$$= \int\int\int \left[ \tilde{k}_{P_X}(x, x')\tilde{l}_{P_Y}(y, y')\tilde{m}_{P_Z}(z, z') \right]$$

$$dP_X(x)dQ(y, z)dQ'(x', y', z')$$

$$= \int\int \underbrace{\int \tilde{k}_{P_X}(x, x')dP_X(x)}_{=\left[\mu_{\tilde{k}_{P_X}}(P_X)\right](x')=0} \tilde{l}_{P_Y}(y, y')\tilde{m}_{P_Z}(z, z')$$

$$dQ(y, z)dQ'(x', y', z')$$

$$= 0.$$

Therefore,

$$\|\Delta_L P\|^2_{\tilde{k}_{P_X} \otimes \tilde{l}_{P_Y} \otimes \tilde{m}_{P_Z}} = \|P_{XYZ}\|^2_{\tilde{k}_{P_X} \otimes \tilde{l}_{P_Y} \otimes \tilde{m}_{P_Z}}$$

$$= \|\Delta_L P\|^2_{k\otimes l\otimes m}.$$

The above is true for any joint distribution $P_{XYZ}$, and in particular for the empirical joint, whereby:

$$\left\|\Delta_L \hat{P}\right\|^2_{k\otimes l\otimes m} = \left\|\hat{P}_{XYZ}\right\|^2_{\tilde{k}_{\hat{P}_X} \otimes \tilde{l}_{\hat{P}_Y} \otimes \tilde{m}_{\hat{P}_Z}}$$

$$= \frac{1}{n^2}\left(\tilde{K} \circ \tilde{L} \circ \tilde{M}\right)_{++}.$$

## A.3 Proof of Proposition 4

Consider the element of $\mathcal{H}_k \otimes \mathcal{H}_l \otimes \mathcal{H}_m$ given by $\mathbb{E}_{XYZ}k(\cdot, X) \otimes l(\cdot, Y) \otimes m(\cdot, Z)$. This can be identified with a Hilbert-Schmidt uncentered covariance operator $C_{(XY)Z} : \mathcal{H}_k \otimes \mathcal{H}_l \to \mathcal{H}_m$, such that $\forall f \in \mathcal{H}_k, g \in \mathcal{H}_l, h \in \mathcal{H}_m$:

$$\langle C_{(XY)Z}[f \otimes g], h\rangle_{\mathcal{H}_m} = \mathbb{E}_{XYZ}f(X)g(Y)h(Z).$$

Table 3: V-statistics for various hypotheses

| hypothesis | V-statistic | hypothesis | V-statistic |
|---|---|---|---|
| $(X, Y) \perp\!\!\!\perp Z$ | $\frac{1}{n^2}\left(K \circ L \circ \tilde{M}\right)_{++}$ | $\Delta_{(X)} P = 0$ | $\frac{1}{n^2}\left(K \circ \tilde{L} \circ \tilde{M}\right)_{++}$ |
| $(X, Z) \perp\!\!\!\perp Y$ | $\frac{1}{n^2}\left(K \circ \tilde{L} \circ M\right)_{++}$ | $\Delta_{(Y)} P = 0$ | $\frac{1}{n^2}\left(\tilde{K} \circ L \circ \tilde{M}\right)_{++}$ |
| $(Y, Z) \perp\!\!\!\perp X$ | $\frac{1}{n^2}\left(\tilde{K} \circ L \circ M\right)_{++}$ | $\Delta_{(Z)} P = 0$ | $\frac{1}{n^2}\left(\tilde{K} \circ \tilde{L} \circ M\right)_{++}$ |
| | | $\Delta_L P = 0$ | $\frac{1}{n^2}\left(\tilde{K} \circ \tilde{L} \circ \tilde{M}\right)_{++}$ |

By replacing $k, l, m$ with kernels centered at the marginals, we obtain a centered covariance operator $\Sigma_{(XY)Z}$, for which

$$
\begin{aligned}
\left\langle \Sigma_{(XY)Z}\left[f \otimes g\right], h \right\rangle_{\mathcal{H}_m} &= \mathbb{E}_{XYZ}\tilde{f}(X)\tilde{g}(Y)\tilde{h}(Z) \\
&= \operatorname{cov}\left[f(X), g(Y), h(Z)\right],
\end{aligned}
$$

where we wrote $\tilde{f}(X) = f(X) - \mathbb{E}f(X)$, and similarly for $\tilde{g}$ and $\tilde{h}$. Using the usual isometries between Hilbert-Schmidt spaces and the tensor product spaces:

$$
\begin{aligned}
&\left\| \Sigma_{(XY)Z} \right\|_{HS}^2 \\
={} &\left\| \mathbb{E}_{XYZ}\tilde{k}_{P_X}(\cdot, X) \otimes \tilde{l}_{P_Y}(\cdot, Y) \otimes \tilde{m}_{P_Z}(\cdot, Z) \right\|_{\mathcal{H}_k \otimes \mathcal{H}_l \otimes \mathcal{H}_m}^2 \\
={} &\| P_{XYZ} \|_{\tilde{k}_{P_X} \otimes \tilde{l}_{P_Y} \otimes \tilde{m}_{P_Z}}^2 \\
={} &\| \Delta_L P \|_{k \otimes l \otimes m}^2 .
\end{aligned}
$$

Now, consider the supremum of the three-way covariance taken over the unit balls of respective RKHSs:

$$
\begin{aligned}
\sup_{f,g,h} \operatorname{cov}\left[f(X), g(Y), h(Z)\right] &= \sup_{f,g,h} \left\langle \Sigma_{(XY)Z}\left[f \otimes g\right], h \right\rangle_{\mathcal{H}_m} \\
&= \sup_{f,g} \left\| \Sigma_{(XY)Z}\left[f \otimes g\right] \right\|_{\mathcal{H}_m} \\
&\leq \sup_{F \in \mathcal{H}_k \otimes \mathcal{H}_l} \left\| \Sigma_{(XY)Z} F \right\|_{\mathcal{H}_m} \\
&= \left\| \Sigma_{(XY)Z} \right\|_{op} \leq \left\| \Sigma_{(XY)Z} \right\|_{HS} .
\end{aligned}
$$

and thus, $\| \Delta_L P \|_{k \otimes l \otimes m} = 0$ implies $\sup_{f,g,h} \operatorname{cov}\left[f(X), g(Y), h(Z)\right] = 0$. Conversely, if $\operatorname{cov}\left[f(X), g(Y), h(Z)\right] = 0 \, \forall f, g, h$, then $\Sigma_{(XY)Z}\left[f \otimes g\right] \equiv 0 \, \forall f, g$, so the linear operator $\Sigma_{(XY)Z}$ vanishes.

## B   The effect of centering

In a two-variable test, either or both of the kernel matrices can be centered when computing the test statistic since $\left(K \circ \tilde{L}\right)_{++} = \left(\tilde{K} \circ L\right)_{++} = \left(\tilde{K} \circ \tilde{L}\right)_{++}$. To see this, simply note that by the idempotence of $H$,

$$
\begin{aligned}
\left(K \circ \tilde{L}\right)_{++} &= tr(KHLH) \\
&= tr(KH^2LH^2) \\
&= tr(HKH^2LH) \\
&= (HKH \circ HLH)_{++} \\
&= \left(\tilde{K} \circ \tilde{L}\right)_{++} . 
\end{aligned} \tag{7}
$$

Table 4: An example of Lancaster interaction measure vanishing for the case where neither variable is independent of the other two.

| $P(0,0,0) = 0.2$ | $P(0,0,1) = 0.1$ |
|---|---|
| $P(0,1,0) = 0.1$ | $P(0,1,1) = 0.1$ |
| $P(1,0,0) = 0.1$ | $P(1,0,1) = 0.1$ |
| $P(1,1,0) = 0.1$ | $P(1,1,1) = 0.2$ |

This is no longer true in the three-variable case, where centering of each matrix has a different meaning. Various hypotheses and their corresponding V-statistics are summarized in Table 3. Note that the "composite" hypotheses are obtained simply by an appropriate centering of Gram matrices.

## C   $\Delta_L P = 0 \nRightarrow (X,Y) \perp\!\!\!\perp Z \vee (X,Z) \perp\!\!\!\perp Y \vee (Y,Z) \perp\!\!\!\perp X.$

Consider the following simple example with binary variables $X$, $Y$, $Z$ with the $2 \times 2 \times 2$ probability table given in Table 4. It is readily checked that all conditional covariances are equal, so $\Delta_L P = 0$. It is also clear, however, that neither variable is independent of the other two. Therefore, a test for Lancaster interaction *per se* is not equivalent to testing for the possibility of any factorization of the joint distribution, but our empirical results suggest that it can nonetheless provide a useful surrogate. In other words, while rejection of the null hypothesis $\Delta_L P = 0$ is highly informative and implies that interaction is present and *no* non-trivial factorization of the joint distribution is available, the acceptance of the null hypothesis should be considered carefully and additional methods to rule out interaction should be sought.

## D   Permutation test

A permutation test for total independence is easy to construct: it suffices to compute the value of the statistic (either the Lancaster statistic $\left\| \Delta_L \hat{P} \right\|_{k \otimes l \otimes m}^2$ or the total independence statistic $\left\| \Delta_{tot} \hat{P} \right\|_{k \otimes l \otimes m}^2$) on $\left\{ \left( X^{(i)}, Y^{(\sigma i)}, Z^{(\tau i)} \right) \right\}_{i=1}^n$, for randomly drawn independent permutations $\sigma, \tau \in S_n$ in order to obtain a sample from the null distribution.

When testing for *only one* of the hypotheses $(Y,Z) \perp\!\!\!\perp X$, $(X,Z) \perp\!\!\!\perp Y$, or $(X,Y) \perp\!\!\!\perp Z$, either with a Lancaster statistic or with a standard two-variable kernel statistic, only one of the samples should be permuted, e.g., if testing for $(Y,Z) \perp\!\!\!\perp X$, statistics should be computed on $\left\{ \left( X^{(\sigma i)}, Y^{(i)}, Z^{(i)} \right) \right\}_{i=1}^n$, for $\sigma \in S_n$. However, when testing for the disjunction of these hypotheses, i.e., for the existence of a nontrivial factorization of the joint distribution, we are within a multiple hypothesis testing framework (even though one may deal with a single test statistic, as in the Lancaster case). To ensure that the required confidence level $\alpha = 0.05$ is reached for the factorization hypothesis, in the experiments reported in Figure 3, the Holm's sequentially rejective Bonferroni method [35] is used for both the two-variable based and for the Lancaster based factorization tests. Namely, $p$-values are computed for each of the hypotheses $(Y,Z) \perp\!\!\!\perp X$, $(X,Z) \perp\!\!\!\perp Y$, or $(X,Y) \perp\!\!\!\perp Z$ using the permutation test, and sorted in the ascending order $p_{(1)}, p_{(2)}, p_{(3)}$. Hypotheses are then rejected sequentially if $p_{(l)} < \frac{\alpha}{4-l}$. The factorization hypothesis is then rejected if and only if all three hypotheses are rejected.

## E   Asymptotic behavior

Using terminology from [26], kernels $k$ and $k'$ are said to be equivalent if they induce the same semimetric on the domain, i.e., $k(x,x) + k(x',x') - 2k(x,x') = k'(x,x) + k'(x',x') - 2k'(x,x')$ $\forall x, x'$. It can be shown that the Lancaster statistic is invariant to changing kernels within the kernel equivalence class, i.e., that

$$\left\| \Delta_L \hat{P} \right\|_{k \otimes l \otimes m}^2 = \left\| \Delta_L \hat{P} \right\|_{k' \otimes l' \otimes m'}^2,$$

whenever $k, k', l, l'$ and $m, m'$ are equivalent pairs. From here,

$$\left\|\Delta_L \hat{P}\right\|^2_{k \otimes l \otimes m} = \left\|\Delta_L \hat{P}\right\|^2_{\tilde{k}_{P_X} \otimes \tilde{l}_{P_Y} \otimes \tilde{m}_{P_Z}}.$$

In Section A.2, we were able to show a similar expression but only for changing $k$ to its version $\tilde{k}_{\hat{P}_X}$ centered at the *empirical marginal*. Now, under the assumption of total independence, i.e., that $P_{XYZ} = P_X P_Y P_Z$, the dominating term in $\left\|\Delta_L \hat{P}\right\|^2_{\tilde{k}_{P_X} \otimes \tilde{l}_{P_Y} \otimes \tilde{m}_{P_Z}}$ is $\left\|\hat{P}_{XYZ}\right\|^2_{\tilde{k}_{P_X} \otimes \tilde{l}_{P_Y} \otimes \tilde{m}_{P_Z}}$. By standard arguments, under total independence, this converges in distribution to a sum of independent chi-squared variables,

$$n\left\|\hat{P}_{XYZ}\right\|^2_{\tilde{k}_{P_X} \otimes \tilde{l}_{P_Y} \otimes \tilde{m}_{P_Z}} \rightsquigarrow \sum_{a=1}^{\infty}\sum_{b=1}^{\infty}\sum_{c=1}^{\infty} \lambda_a \eta_b \theta_c N_{abc}^2, \tag{8}$$

where $\{\lambda_a\}$, $\{\eta_b\}$, $\{\theta_c\}$ are, respectively, eigenvalues of integral operators associated to $\tilde{k}_{P_X}$, $\tilde{l}_{P_Y}$ and $\tilde{m}_{P_Z}$, and $N_{abc} \overset{i.i.d.}{\sim} \mathcal{N}(0,1)$. Other terms in $\left\|\Delta_L \hat{P}\right\|^2_{\tilde{k}_{P_X} \otimes \tilde{l}_{P_Y} \otimes \tilde{m}_{P_Z}}$ can be shown to drop to zero at a faster rate, as in the two-variable case. The resulting distribution of such a sum of chi-squares can, in principle, be estimated using a Monte Carlo method, by computing a number of eigenvalues of $\tilde{K}$, $\tilde{L}$ and $\tilde{M}$, as in [36, 18]. This is of little practical value though, as it is in most cases simpler and faster to run a permutation test, as we describe in Appendix D. On the other hand, the above result quantifies the highest order of bias of the V-statistic under total independence to be equal to $\frac{1}{n}\sum_{a=1}^{\infty} \lambda_a \sum_{b=1}^{\infty} \eta_b \sum_{c=1}^{\infty} \theta_c$, which can be estimated as $\frac{1}{n^4}Tr(\tilde{K})Tr(\tilde{L})Tr(\tilde{M})$. We emphasize that (8) refers to a *null distribution under total independence* - if say, the null holds because $(X, Y) \perp\!\!\!\perp Z$, but $X$ and $Y$ are dependent, one needs to instead consider a kernel on $\mathcal{X} \times \mathcal{Y}$ centered at $P_{XY}$ and the eigenvalues of its integral operator then replace $\{\lambda_a \eta_b\}$ (triple sum becomes a double sum). This also implies that the bias term needs to be corrected appropriately.

# F Some useful basic matrix algebra

**Lemma 7.** *Let $A$, $B$ be $n \times n$ matrices. The following results hold:*

1. $\mathbf{1}^\top \mathbf{1} = n$

2. $[\mathbf{11}^\top]_{ij} = 1$, $\forall i, j$, *and thus* $\left(\mathbf{11}^\top\right)_{++} = n^2$

3. $\left(I - \frac{1}{n}\mathbf{11}^\top\right)^2 = I - \frac{1}{n}\mathbf{11}^\top$.

4. $[A\mathbf{1}]_i = A_{i+}$, $\left[\mathbf{1}^\top A\right]_j = A_{+j}$

5. $\mathbf{1}^\top A\mathbf{1} = A_{++}$

6. $\left(A\mathbf{11}^\top\right)_{++} = \left(\mathbf{11}^\top A\right)_{++} = nA_{++}$

7. $(\alpha A + \beta B)_{++} = \alpha A_{++} + \beta B_{++}$

8. $\left(A\mathbf{11}^\top B\right)_{++} = A_{++}B_{++}$.

*Proof.* (3):

$$\left(I - \frac{1}{n}\mathbf{11}^\top\right)^2 = I - \frac{2}{n}\mathbf{11}^\top + \frac{1}{n^2}\mathbf{1}\underbrace{\mathbf{1}^\top \mathbf{1}}_{n}\mathbf{1}^\top.$$

(8): From (4), $\left[A\mathbf{11}^\top B\right]_{ij} = A_{i+}B_{+j}$, implying

$$\left(A\mathbf{11}^\top B\right)_{++} = \sum_{i=1}^{n} A_{i+} \sum_{j=1}^{n} B_{+j} = A_{++}B_{++}.$$

$\square$

Now, let $K$ be a symmetric matrix, and denote $H = I - \frac{1}{n}\mathbf{1}\mathbf{1}^\top$ (the centering matrix). Then:

$$
\begin{aligned}
HKH &= \left(I - \frac{1}{n}\mathbf{1}\mathbf{1}^\top\right) K \left(I - \frac{1}{n}\mathbf{1}\mathbf{1}^\top\right) \\
&= K - \frac{1}{n}\left(K_+ + K_+^\top\right) + \frac{1}{n^2}K_{++}\mathbf{1}\mathbf{1}^\top.
\end{aligned}
$$

Note that:

$$
\begin{aligned}
(HKH)_{++} &= K_{++} - \frac{1}{n}\left((K_+)_{++} + \left(K_+^\top\right)_{++}\right) + \frac{1}{n^2}K_{++}\left(\mathbf{1}\mathbf{1}^\top\right)_{++} \\
&= K_{++} - 2K_{++} + K_{++} = 0.
\end{aligned}
$$

**Lemma 8.** *The following results hold:*

1. $A \circ \mathbf{1}\mathbf{1}^\top = \mathbf{1}\mathbf{1}^\top \circ A = A$

2. $(I \circ A)_{++} = tr(A)$

3. $(A \circ B)_{++} = tr(AB^\top)$

4. For a symmetric matrix $K$ and any matrix $A$, $(A \circ K_+)_{++} = (AK)_{++}$, $\left(A \circ K_+^\top\right)_{++} = (KA)_{++}$

5. For symmetric matrices $K$, $L$, $(K_+ \circ L_+)_{++} = \left(K_+^\top \circ L_+^\top\right)_{++} = n\,(KL)_{++}$

6. For symmetric matrices $K$, $L$, $\left(K_+ \circ L_+^\top\right)_{++} = \left(K_+^\top \circ L_+\right)_{++} = K_{++}L_{++}$.

*Proof.* (4):$(A \circ K_+)_{++} = tr\left(AK\mathbf{1}\mathbf{1}^\top\right) = \left(AK \circ \mathbf{1}\mathbf{1}^\top\right)_{++} = (AK)_{++}$. (5): $(K_+ \circ L_+)_{++} = (K_+L)_{++} = \left(\mathbf{1}\mathbf{1}^\top KL\right)_{++} = n\,(KL)_{++}$. □

**Proposition 9.** *Denote $H = I - \frac{1}{n}\mathbf{1}\mathbf{1}^\top$. Then:*

$$
(K \circ HLH)_{++} = (K \circ L)_{++} - \frac{2}{n}(KL)_{++} + \frac{1}{n^2}K_{++}L_{++}.
$$

*Proof.* Let $K$ and $L$ be symmetric matrices and consider $K \circ HLH$. We obtain:

$$
\begin{aligned}
K \circ HLH &= K \circ \left(L - \frac{1}{n}\left(L_+ + L_+^\top\right) + \frac{1}{n^2}L_{++}\mathbf{1}\mathbf{1}^\top\right) \\
&= K \circ L - \frac{1}{n}\left(K \circ L_+ + K \circ L_+^\top\right) + \frac{1}{n^2}L_{++}K,
\end{aligned}
$$

so that:

$$
(K \circ HLH)_{++} = (K \circ L)_{++} - \frac{2}{n}(KL)_{++} + \frac{1}{n^2}K_{++}L_{++}.
$$

□

**Corollary 10.** $tr(HLH) = tr(L) - \frac{1}{n}L_{++}$