[Reviews · NeurIPS 2013]

Submitted by Assigned_Reviewer_5

The paper extends the existing machine of kernel dependence tests to three way interactions. This is an interesting and well executed piece of work that attacks an important problem for which there are very limited tools at present. The paper is generally well written.

Some minor comments.
Explain V-structure more explicitly on its first use.
Give a pointer to V-statistics (I had heard of U-Statistics, but not V-statistics)
Equation (1) does not make sense as written. You need to define P^* more clearly. The "where" after (1) does not make it clear.

In the experiments, explain sample size.
Summary: Nice extension of kernel dependence tests to three-way interactions.

Submitted by Assigned_Reviewer_6

This paper provides a very interesting nonparametric approach to test for three-variable interactions (and independence) using embedding properties of ISPD kernels. The paper is overall well-written, clear and exhaustive in its coverage of the underlying algebraic proofs. With the need for efficient statistical testing tools to detect complex interaction effects in bioinformatics, this work seems promising in its applications.
Given the stated illustrative purpose of identifying gene interactions, it could have been nice to see experiments on non-synthetic data and/or comparison of performances to other genomics methods (such as based on partial correlation). Also, the possibility raised in the concluding paragraph that the test could be extended to more than 3 variables seems to be somewhat contradicted by the point raised in section 4.3. Finally, it would be of great interest to see how the method behaves in large-dimensional spaces.
Minor issue: There is no reference to Figure 3 in the main text
Summary: A solid and novel statistical approach, with promising applications. Would benefit from more exhaustive performance comparison to other state-of-the-art methods for interaction testing

Submitted by Assigned_Reviewer_7

This paper addresses a problem of building hypothesis tests based on kernel methods for identifying three variable interactions. This work extends HSIC (Hilbert Schmidt Independence Criterion), which is designed to detect two variable interactions based on kernel methods. There exists several types of three variable interactions, but the one authors try to detect one called Lancaster interaction, which is a singed measure defined as ¥delta L = Pxyx -PxyPz - PxzPy - PyzPx - 2 PxPyPz. In the proposed procedure, the above marginal and joint probabilities are all embedded into kernel Hilbert space, and guaranteed to work as long as they satisfy ISPD (Integrally Strictly Positive Definite) condition. In simulation experiments, permutation test is employed with ¥delta L as a test statistic to show the ability of the proposed method in identifying three way interactions.

In the middle of the page 8, "Figure 2 plots the Type II error" should be "Figure 3".
Summary: The paper is clearly written with appropriate references and proofs. As is written in the manuscript, extension of this method to detecting higher degrees of interactions, and to detecting structured interactions is useful.
Author Feedback

Author rebuttal: We thank the reviewers for their kind assessment of our paper. We are happy to incorporate the corrections and suggestions made by the reviewers to improve the clarity of our presentation. Regarding the suggestions of reviewer 2: we certainly hope our method will find application in bioinformatics and other areas; this is a route we are pursuing, and we have prepared downloadable software. We also take the reviewer's point that our conclusion was at odds with our discussion of the difficulty of measuring four-way interactions or higher, and have revised accordingly.